# Unmasking the Adverse Impacts of Sex Bias on Science and Research Animal Welfare

**DOI:** 10.3390/ani13172792

**Published:** 2023-09-02

**Authors:** Elizabeth A. Nunamaker, Patricia V. Turner

**Affiliations:** 1Global Animal Welfare and Training, Charles River Laboratories, Wilmington, MA 01887, USA; elizabeth.nunamaker@crl.com; 2Department of Pathobiology, University of Guelph, Guelph, ON N1G 2W1, Canada

**Keywords:** sex bias, animal welfare, biomedical research, reproducibility, translatability, drug development

## Abstract

**Simple Summary:**

Sex bias—the use of one sex over the other—is a common practice in biomedical research, typically with over selection of male animals. There are a number of reasons that this practice is common, but it has resulted in dosing errors and unintended side effects in a number of cases in women when products are given without adequate testing in female animals. Sex bias can also result in animal welfare issues given that both sexes are born in approximately equal numbers. Welfare issues include overproduction of the unwanted sex, inadequate ability to recognize and treat pain in female animals, stress associated with differential housing needs based on animal sex, and potential wastage of animals if study results are incorrect or studies need to be redone because only one sex was studied. Even though many government agencies and funding sources now require both sexes to be studied in biomedical research, single-sex studies are still common. More systematic planning and reporting of study details is needed, as well as exploring sex selection technology used in other animal production sectors when single-sex studies are justified, to reduce animal waste.

**Abstract:**

Sex bias in biomedical and natural science research has been prevalent for decades. In many cases, the female estrous cycle was thought to be too complex an issue to model for, and it was thought to be simpler to only use males in studies. At times, particularly when studying efficacy and safety of new therapeutics, this sex bias has resulted in over- and under-medication with associated deleterious side effects in women. Many sex differences have been recognized that are unrelated to hormonal variation occurring during the estrous cycle. Sex bias also creates animal welfare challenges related to animal over-production and wastage, insufficient consideration of welfare (and scientific) impact related to differential housing of male vs female animals within research facilities, and a lack of understanding regarding differential requirements for pain recognition and alleviation in male versus female animals. Although many funding and government agencies require both sexes to be studied in biomedical research, many disparities remain in practice. This requires further enforcement of expectations by the Institutional Animal Care and Use Committee when reviewing protocols, research groups when writing grants, planning studies, and conducting research, and scientific journals and reviewers to ensure that sex bias policies are enforced.

## 1. Introduction

An historical overreliance on male animals in the drug development process has resulted in women taking drugs at inappropriate doses and experiencing side effects rarely reported in men. This has required the FDA to reevaluate safety and efficacy study results to (re)establish differential dosing for men and women after the drugs were approved [1,2]. Many FDA-approved therapeutics have elevated blood levels and longer elimination times in women, as well as being associated with a higher incidence of adverse drug reactions [3]. Had both sexes of animals been used in the preclinical testing phase, the sex-related differences in drug metabolism and clearance may have been identified sooner, and many adverse drug reactions in women may have been avoided [2]. 

This sex bias in biomedical research has been well documented. In a study across 10 fields of biology, it was found that 80% of the animals used were male [4]. It was further found that, even when both sexes were used, only one-third of studies analyzed the results by sex [4]. While there are many different justifications used to propagate sex bias in biomedical research [5], appropriate experimental design should be used to effectively accommodate both sexes, maintain or increase power, and avoid interpretation errors and associated increased costs [6,7,8]. Beyond experimental and human health concerns, sex bias also represents an animal welfare concern. Specifically, there is evidence to suggest that sex bias results in overproduction of research animals (see Section 4.1), inadequate pain management (see Section 4.2), and significant animal wastage (see Section 4.4).

The ubiquity of sex bias in biomedical research has developed over time into the current situation. In this paper, we discuss how sex bias developed over the past 100 years and the ongoing efforts to address it. The underlying justifications used, scientific and not, are explored, and the implications for animal welfare are described. Lastly, specific methods to minimize sex bias in biomedical studies are suggested.

## 2. How Did Sex Bias Develop and How Pervasive Is It?

Sex-based differences in animal studies are not a revolutionary concept. Differences in biological responses have been documented in the literature since the 1930s [9,10,11,12]. Early studies were focused on the increased variability in learning behavior of female rats. The perceived higher variability of responses to experimental challenges in females and the differences between the sexes provided a justification for studying only male animals as a means of simplification. This was further compounded by prominent literature in the 1960s that encouraged investigators to keep animal numbers low and reduce experimental variability by specifically using a single sex [13]. By the late 1970s, the literature was flooded with studies demonstrating a sex difference in many physiological systems [14]. Instead of using this as justification to study both sexes, the rationale for studying only males became entrenched in animal-based research studies [15].

An awareness about the limitations of the pervasive bias towards the use of male animals first became apparent in the 1990s [16,17,18]. Sex bias gained considerable interest between 1997 and 2000, when the US Food and Drug Administration suspended distribution and sales of eight different prescription drugs due to severe adverse effects that were reported in women taking them [19]. Ultimately, the root cause of the suspensions was systemic sex bias in the drug development process resulting in dose recommendation errors. The compounds were initially screened using cell cultures of male origin, preclinical testing was performed in male animals only, and clinical testing was primarily completed in men [19]. These events collectively sparked the interest in studying the sex and gender bias that is still seen today in both biomedical and clinical research.

In 2010, Beery and Zucker conducted a survey of animal use in neuroscience and biomedical research and found that, after 20 years of awareness of sex bias, the practice continued to be pervasive in animal studies. Specifically, they found a male bias in 8 of 10 surveyed disciplines, with single-sex studies of male animals outnumbering those of females, 5.5 to 1 [4]. In response to this demonstrated male bias, the National Institutes of Health began requiring that sex be included as an experimental variable in grant applications [20,21,22]. Other funding agencies subsequently followed their lead. Despite these efforts, the scientific literature continues to be full of examples of male sex bias in animal research [5], especially in the fields of pain management [23], cardiovascular disease [24,25], diabetes mellitus [25], alcohol-related diseases [26], and development of surgical methods [27]. Potentially equally damaging to scientific rigor and reproducibility, it was also common to not report sex at all during this time, as occurred in upwards of 25% of published animals studies and 76% of cell culture studies [27].

In 2020, when a 10-year follow-up study was conducted on sex reporting [5], there was some evidence of improvement, but sex bias remained pervasive. There was an increase in the proportion of studies that included both sexes, but no change in the proportion of studies that included data analyzed by sex. Most studies continued to fail to provide a rationale for the use of animals of a single sex in their studies. Further, there was also a lack of sex-based analyses, and those that conduct a sex-based analysis relied on misconceptions surrounding the hormonal variability of females. This data suggests that there is still significant work to be done in experimental design and data analysis to include both sexes.

## 3. Why One Sex May Be Preferred in Research Settings

There are several reasons why one sex may be preferred over the other in a research setting [8]. Some of these have been used historically to propagate the sex bias seen today, while other reasons are legitimate justification for the use of a single sex. Woitowich and colleagues captured and consolidated the justifications found in the literature into six themes: 30% known sex difference or sex effect, 27% increased experimental variability, 13% experimental conditions, 13% limited sample size, 10% inability to sex subjects, and 7% issues with animal husbandry [5]. These are each discussed below.

### 3.1. Known Sex Differences or Sex Effects on Research

There are diseases and physiologic processes that occur in only one sex. There are many known sex-linked traits and conditions, such as hemophilia A, Duchenne muscular dystrophy, Fragile X syndrome, breast cancer, conception, and in utero fetal development. The disease or physiologic process of interest inherently limits the ability to study the disease in both sexes. However, it is also critical to understand the role of sex in fundamental physiology and diseases processes, making it important to include both sexes to compare and find these sex effects whenever possible [23,28]. Likewise, when studying safety and efficacy of therapeutics, it’s important to ensure that there are no sex-based differences in treatment safety or efficacy.

### 3.2. Increased Experimental Variability

Females have been long excluded from studies due to misconceptions about their estrous (or menstrual) cycle increasing day-to-day experimental variability and because including them results in a need for more research animals. However, the estrous cycle is typically not a variable that contributes significantly to experimental variability [29,30,31,32,33,34]. Empirical research across multiple rodent species and traits demonstrates that females are not more variable than males, and that for most traits, female estrous cyclicity need not be considered [8]. Even when the estrous cycle is a known or significant variable, experiments can be designed around it, and hormone variation can be incorporated into the design to account for that variability [29,30,31,32,33]. Successful incorporation of hormone variation in study design has previously been documented [30,32,35,36]. It is also noteworthy that individual variation has been documented to be a larger source of variability in behavior than estrous state [37]. 

Practically speaking, a good approach is to compare males with two or more groups of females where the stage of the ovarian cycle is known. A three-group design in mice or rats, for example, could compare males with females on two specific days of the estrous cycle. Alternatively, a five-group design would compare males with females on each of the four days of the mouse or rat estrous cycle [31]. The later design would allow for detection of a sex difference, specifically isolated to a precise day of the estrous cycle.

### 3.3. Experimental Condition

The experimental condition may have inherent sex differences or existing biases that make it difficult to include both sexes. Examples of this include situations such as not including females because collecting vaginal smears to control for stage of estrous cycle adds stress to the animal experience or only using adult females for a behavior study because adult males and juveniles of either sex rarely vocalize [5]. Neither of these examples are strong justifications and push-back would be warranted. A more reasonable justification to exclude one sex would be in the case of studying uterine tumors, which cannot occur in the male sex, or prostate cancer, which does not occur in females. The justification for excluding one sex based on experimental condition should be closely scrutinized to ensure that the reasoning is scientifically sound.

### 3.4. Limited Sample Size

Sample size limitations can make it difficult to analyze data for sex-based effects. These limitations may be due to either having a limited resource (e.g. small population of unique animals) or costs. Potentially increasing the number of animals in a study is a concern due to the associated costs; however, per the Guide for the Care and Use of Laboratory Animals, cost is not an acceptable justification for reduction of animal numbers [38]. A better approach to eliminating the sex bias implications of limited sample size is by using factorial designs to reduce the need for additional animals while including both males and females [8].

### 3.5. Inability to Sex Subjects

There are times when the sex of the animals or tissues being used is not obvious or is truly unknown. This may be the case when working with embryos or slaughterhouse tissues. However, the sex of these tissues can be determined through the use of various molecular techniques, including polymerase chain reaction (PCR), gas chromatography-mass spectrometry (GC-MS), high performance liquid chromatography/mass spectrometry (HPLC-MS/MS), and enzyme-linked immunosorbent assay (ELISA) on those tissues [39,40,41,42,43]. 

### 3.6. Animal Husbandry

Animal husbandry limitations can unknowingly contribute to sex bias in research. This can be due to limited vivarium space, sex-based aggression [44,45], response to husbandry procedures [46], and relative ease of social housing [45] (see Section 4.3 below). To avoid unscheduled breeding and unanticipated offspring, males and females are typically housed in single-sex groups. The ease of social housing can vary by species and by animal age. Aggression between same-sex conspecifics can be a significant welfare concern leading to the use of housing strategies to meet the local or national regulatory requirements, as well as the limitations of the vivarium size (see 4.3 below for a further discussion). An investigator may opt to use one sex during fetal or neonatal development stages for part of a study while using juveniles and adults for other portions of a study with the goal of simplifying the husbandry. Unfortunately, this can create unintended sex bias in a study. Identifying husbandry effects or limitations and effectively preparing for them can prevent this unintended source of sex bias from animal studies.

## 4. The Impact of Sex Bias on Animal Welfare

There are many examples demonstrating how sex bias in biomedical research can impact animal welfare. In this section, we will highlight how sex bias can result in overproduction of research animals, result in inadequate pain management, ignore underlying differences in male versus female stress responses and physiology, and result in animal wastage and poorly reproducible and translatable results.

### 4.1. Overproduction of Research Animals: Ethics and Sex Bias

Minimizing animal waste is an important component of reduction, one of the 3R’s tenets. There is significant and continued interest within the laboratory animal community in reducing surplus animals produced for biomedical research [47]. It has been estimated that >110 million mice and rats are used in science and education each year, although one recent estimate suggests that >110 million mice and rats are used in research each year in the US alone [48,49,50,51,52,53]. Conservatively, and on average, at least 30% overproduction exists in a given colony, even with the most efficient breeding methods in commercial settings [54]. For the EU, this results in an estimated 12.5 million animal surplus; for the UK, there is an excess of at least 1.6 million mice and rats produced [55]. Together, these suggest that there could be overproduction of 25 million or more mice and rats worldwide. Some of these unneeded animals may be used for training, harvesting tissues and fluids for subsequent research, or humanely killed and donated or sold for animal feed; however, many can only be incinerated after killing because of strict regulations surrounding disposal of genetically engineered animals worldwide [56]. Some countries have tried to address the ethical concerns created by animal overproduction. In 2022, in an effort to reduce unwanted male dairy calves and male layer chicks produced on farms, German legislation was enacted that makes it illegal to kill surplus production animals without cause, including animals produced for biomedical research [57]. This approach may have the unintended consequence of driving research animal production and, ultimately, biomedical science to countries or regions with less rigorous animal welfare standards in an effort to minimize the significant financial and resource burdens of providing lifetime care to unwanted and unneeded research animals.

There are numerous reasons underlying surplus production of animals for biomedical research; however, sex bias is a significant contributor for smaller species, including rodents and rabbits (see Table 1). This can be exacerbated by age and weight restrictions for a given experiment, animal order, or assay. For example, traditional pertussis vaccine potency tests in mice have required all mice in a given group to vary by no more than 4g in body weight, driving the use of one sex because of significant body weight dimorphism in mice, and resulting in significant ordering wastage due to body weight gain variation between the time when animals arrive in the facility and when they can be studied [58]. Some have suggested using mixed sex groups for vaccine potency and challenge trials; however, this is not practical for the majority of assays in which adult animals are used in studies with a duration of three or more weeks [59] because of the risk of unwanted pregnancies. A better approach when these assays must be conducted in mice is to challenge why such tight weight ranges are required when they don’t exist in the human population to whom the vaccines are administered. Broader weight range acceptability would permit animals of both sexes to be used in these assays. Annually, hundreds of thousands of mice are still used for vaccine potency testing, so the impact of this consideration is not insignificant.

When considering stock and inbred strains, sex bias generally leads to an overproduction of female rats and male mice and rabbits [60,61]. Sex bias is a less of an issue for purpose-bred large animal species, including dogs, primates, and pigs since, within biomedical research, the majority are used for toxicology studies that often require equal numbers of male and female animals [60]. It is more difficult to estimate the impact of sex bias for genetically engineered rodent colonies as there may be a large breeding surplus related to a specifically desired and restricted genotype or adverse phenotype that requires colonies to be maintained largely as heterozygotes [62]. The examples in Figure 1 below demonstrate how sex bias may contribute to overproduction of rodents from inbred, stock, and genetically modified animal colonies. It is important to note that some males and females need to be kept back as replacement breeders or to replace animals unsuitable for study (e.g., with malformations), and these form part of the managed surplus. If the scenario is repeated across many colonies or orders, the overproduction issue becomes significantly magnified.

### 4.2. Pain Recognition and Mitigation in Laboratory Animals

The pain literature, including how pain is modeled, studied, and mitigated in laboratory animals, is fraught with male bias [63]. This has contributed to challenges in identifying pain and managing it appropriately in research animals.

#### 4.2.1. Pain Response by Sex

It has been recognized for many decades that male and female rodents respond differently to acute pain initiated by standard analgesiometry tests, with female rodents generally demonstrating a lower pain threshold to mechanical, hot thermal, chemical, and inflammatory nociception assays (for a review, see [64]). Potential reasons underlying sex differences in pain processing include the potential modulating effect of ovarian hormones on pain-evoked behaviors in females, with hypersensitivity to pain noted in the proestrus and estrus phases of the estrous cycle, that is likely linked to circulating estrogen and testosterone levels [65]. In addition, there are sex differences in neural mediation of pain, neuroimmune modulation of pain, and genetic mediation of pain, in addition to qualitative sex differences in cognitive, social, and environmental factors that modulate pain (reviewed by [63]). In contrast, for complex pain models, such as chronic inflammation and neuropathic pain, there has been unclear evidence for sex differences in rodents in pain perception [64], whereas in women, complex chronic painful conditions, such as irritable bowel syndrome, migraine, diabetic neuropathy, postoperative pain, and fibromyalgia are more commonly reported in women and last longer with a higher pain intensity [63,65]. The lack of concordance in female rodents may be due to insufficient power in study designs to detect sex differences in addition to a lack of recognition and study of biologically different processes for pain signaling between sexes [63]. 

Surprisingly, there has been minimal study of sex differences in pain perception for other animal species, including in research, farm, zoologic, and companion animal settings. This may be because of the expense of these models and the difficulty in achieving sufficient sample sizes of a given species and breed, let alone sex, when enrolling veterinary patients in clinical trials. It may also be due to poor recognition in veterinary medicine of species and sex differences in pain processing and response. For example, pain in cats is poorly recognized by many veterinary practitioners compared to pain in dogs, even for the same procedures [66,67]. It has only been relatively recently that neuter procedures for female cats and dogs (i.e., ovariohysterectomy) have been objectively identified to be more painful and require significantly more analgesia than neuter procedures for male cats and dogs (i.e., castration) [68]. Understanding sex differences in pain sensitivity and response in animals are areas requiring more research to support animal welfare.

#### 4.2.2. Pain Mitigation by Sex

Despite decades of research defining sex differences in response to acute and chronic pain, a recent systematic review and meta-analysis examining potential differences in opioid-induced pain relief by sex in humans found inconclusive findings [69]. This is unlikely to be a result of sex having no effect on opioid-induced analgesia and is more likely to be a result of confounding factors. Women typically have higher fat stores compared to men, resulting in a higher apparent volume of distribution whereas men are typically larger and have faster clearance rates for drugs [70]. Differential metabolism of drugs by hepatic cytochrome P450 isoenzymes by sex may also explain differential responses to analgesic drugs. For example, because hepatic expression of Cyp3A4 is higher in women, the effects of some opioids, such as fentanyl, may be reduced compared to men. Conversely, CYP2D6 has higher expression in males, meaning that codeine and other opioids which are preferentially metabolized by CYP2D6 will have a lesser effect in males [71,72]. In mice and rats, males tend to have more body fat than females of the same species, which may skew the effects of lipophilic opioid analgesics oppositely than for humans [73]. Nonsteroidal anti-inflammatory drugs (NSAIDs) and glucocorticoids are known to have different activities and side effects in humans based on sex, likely due to differences in innate and acquired immune system activity and hormonal fluctuations during ovulation in women as well as differences between male and female NSAID pharmacokinetics and pharmacodynamics, but this is poorly characterized [74]. Minimal information is available in the veterinary or laboratory animal medicine literature about differential NSAID activities and sensitivities within a given species by sex. Certainly, to ensure good animal welfare after painful procedures this topic should be prioritized as an area of research to avoid under- and overdosing animals.

### 4.3. Welfare Impact of Differential Housing by Sex in Biomedical Research

An area that has received insufficient attention is the impact of differential housing of many animal species by sex in research settings (discussed from a different viewpoint in 3.6 above). This is an area that has an important impact on animal welfare as well as reproducibility and translatability of experimental findings. In conventional housing systems used in North America and elsewhere, it is not uncommon for intact, sexually mature males of certain species, including mice, guinea pigs, domestic and mini-pigs, primates, and rabbits to be housed individually because of space restrictions for housing, which generally results in animals being unable to escape agonistic interactions and move away, as would occur under more extensive housing environments. Fighting and wounding can be significant, and males may be routinely housed individually after reaching sexual maturity. In contrast, females of a given species are routinely housed in groups, with offspring, if being held for breeding. For most species, this might somewhat mimic the natural state in which few sexually mature males would be within the same social group; however, these animals would be in constant contact with females and juveniles of the same species rather than living completely solitary lives. These differential housing details are rarely mentioned in published methodology, and yet social housing of vertebrate species is thought to be a critical determinant of individual fitness and health [75]. In addition to inducing states of chronic stress, social isolation may also impact metabolism, biological rhythms, cognition, immunity and inflammation, and oxidative stress and aging in many species, including humans (reviewed by [75]). How this commonly employed differential housing environment impacts male behavior and physiology and whether this approach may skew data when only male animals are used in some types of research are unknown.

Many efforts have been made to try to socially co-house some males of some species to enhance their welfare, but it is impossible to make hard and fast rules for how to do this successfully for all breeds and/or strains of a given species within the constraints of conventional housing [76]. For example, keeping male mice from the same litter together, grouping males prior to sexual maturity, and transferring less heavily soiled nesting material at cage change have been successful methods for keeping some strains of male mice together [77]. Other factors, such as increased cage density and specific strain, were strongly predictive of significant aggression in mice [78], and more work needs to be done to find solutions for compatible housing of male research animals across species [44]. 

### 4.4. Sex Bias and Research Animal Waste

The final example of how sex bias and research animal waste can adversely impact animal welfare and the 3Rs relates to animal waste due to poor reproducibility and translatability of research. In a review of >15,000 biomedical research publications in 2014, only 50% of authors reported animal sex, and when reported, sex bias was noted and varied by preclinical model, with strongest male bias in cardiovascular studies and strongest female bias in infectious disease studies [25]. Beyond initial investigative studies, this sex bias in seeking therapeutic targets and new medicines creates a real risk for misinformation given that female animals are not simply scaled down versions of males [79]). Animal models can only be relevant for both male and female humans when both sexes are used. Single-sex studies may result in animal waste if new test articles are inactive in one sex; they can also result in human safety risks if an agent proves to be more potent in one sex, and this is not identified because of single-sex animal studies. For example, calcitonin gene-related peptide (CGRP) antagonists are of interest for the treatment of migraine in humans. Injection of CGRP triggers migraines in people, and when initially modelled in rodents, poor efficacy was noted in the CGRP-migraine model. Subsequently, when researchers returned to the original studies, it was noted that the testing was conducted exclusively in male rats. When female rats were used instead, very significant improvements were seen when CGRP antagonists were given [80]. Given that migraines occur more often in women, this reinforces the need to use both sexes in animal studies to avoid making incorrect generalizations and to avoid wasting animals in studies.

Not only is including both sexes in experimental design important, but reporting either the existence or the lack of sex difference is also critical to minimizing animal waste. There is a misconception that a finding of no sex difference (a negative result) need not be reported. When both sexes are used, it remains common practice to only report when a difference between sexes is identified. In fact, when a sex difference is identified, half of those studies treat it as a major finding and highlight the finding in the title or abstract [81]. Conversely, in the 44% of published studies in which a sex difference was not specifically found, there is no mention of evaluating the data for a sex-based effect. Sex is not uniformly treated or ignored as a biological variable between scientific fields. The sexes are most commonly compared in endocrinology studies (93%) and least often evaluated in neuroscience studies (33%) [81]. As a result of this practice, experiments may later need to be repeated in both sexes due to the missing information. This potentially contributes to significant animal waste. 

## 5. How Can Sex Bias Be Minimized in Biomedical Research?

Increased awareness of the potential harms of sex bias is an important first step in addressing the issue; however, the scientific community needs tangible and practical solutions to help it overcome the pitfalls of sex bias. There are already several tools available to help guide the scientific planning and reporting practices, and learning how to effectively use these tools can help to propagate excellence in study design, data analysis, and reporting behaviors. Funding agencies reinforce these good behaviors and emphasize minimizing sex bias by increasingly requiring that investigators include both sexes in their research proposals or include strong scientific justification for why they are not needed. Once funded studies are completed, accountability for complete and transparent reporting in scientific reports is key to minimizing sex bias in scientific literature. If the scientific community doesn’t self-govern in this space, some countries may use legislation to minimize sex bias and the associated animal wastage. On top of study design and transparent reporting practices, technology advancements also may be helpful in minimizing animal wastage when a single sex may be legitimately needed.

### 5.1. Awareness and Education

When there is awareness that sex bias exists and the associated welfare harms are identified, refined practices can be taught and preemptively employed to prevent sex bias at each step in the scientific process (Table 2). Mindful elimination of sex bias should follow the PREPARE (Planning Research and Experimental Procedures on Animals: Recommendations for Excellence) Guidelines through to reporting following the ARRIVE (Animal Research: Reporting of In Vivo Experiments) Guidelines [82,83,84].

The path toward minimizing sex bias in animal studies begins with an appropriate and rigorous literature search for sex differences in the targeted area of research interest. Evaluating the resulting search findings is important for developing an understanding of both the strengths and limitations of the current literature and will aid in the development of appropriate design of future studies. There could be known differences between the sexes that may or may not be relevant to the current research question. Critically evaluating the previous supporting work will help to determine if there is an underlying sex difference that needs to be addressed and accommodated. As part of the literature search, the PREPARE Guidelines checklist specifically recommends that researchers: (1) form a clear hypothesis with primary and secondary outcomes; (2) consider the use of systematic reviews; (3) decide upon databases and information specialists to be consulted and construct search terms; (4) assess the relevance of the species to be used, including its biology and suitability to answer the experimental questions with the least suffering and its welfare needs; and (5) assess the reproducibility and translatability of the project. A complete literature search will ultimately help minimize experimental bias, including sex bias, and inappropriate statistical methodology, which are common contributors to poor study design [82].

A well-thought-out study design is key to adequately powering a study to clearly identify and accommodate sex differences [6]. Because many researchers are not trained to do this, it may be important to engage a biostatistician who can assist with the process. Arguably, the most appropriate study design to identify a sex effect is a factorial design [7,8]. This approach reduces the need for additional research subjects while appropriately powering the experiment to identify both the desired experimental effect and any specific difference between the sexes. Factorial design simulations demonstrate that there is no loss of power to detect treatment effects when splitting the sample size across sexes in most scenarios [7]. It may be considered best practice to use a factorial experimental design and split the sample size across both male and female animals.

Once a study has been appropriately designed, using a systematic approach to conduct the experiment will help prevent introducing additional sex bias. One way to accomplish this is to use housing strategies that avoid differential housing of animals based on sex and that account for potential aggression between conspecifics of the same sex [86]. The housing system can introduce sex bias and potentially undermine the most well-designed animal study. As such, there is a need to assess housing effects on research animals and research paradigms to minimize unintended introduction of sex bias into animal studies [44]. Furthermore, blinding observers to the sex of the animals when possible will help to remove any preconceived biases that observer may have. 

Using appropriate data analysis methods, including blinding of analysts to animal sex and treatment group, is important for identifying true positive differences between the sexes. It is noteworthy that finding no sex difference is just as significant as the presence of a difference. Similar to the issues with experimental design, many scientists are not trained on best practices for detecting sex-based difference. As a result, it is common for studies to incorrectly claim a sex difference when there is none, and vice versa [81]. As such, there is a need for continuing efforts to train researchers on how to appropriately test for and report sex differences in their data to promote rigor and reproducibility in biomedical research [7,8,81]. 

The final step in minimizing sex bias is transparent reporting of all aspects of the study. Complete reporting, following the ARRIVE 2.0 Guidelines [83], ensures that a scientist from another institution can accurately recreate the experimental condition and data analysis from the details provided in the manuscript and achieve similar experimental results. Complete and transparent reporting also allows the scientific community to assess study results for sex effects or possible sex bias in the study design or study analysis. Following this entire process from start to finish will help minimize sex bias in animal research and improve reproducibility and translatability of animal-based research. 

### 5.2. Funding Agency Requirements

Funding agencies have recognized that sex is an important biological variable in biomedical research which should be controlled. Many now require researchers to use both sexes in their experiments or clearly justify why they are only using either males or females in their studies. This trend began with the National Institutes of Health [87] announcing a policy aimed at integrating sex as a biological variable (SABV) into biomedical research in 2014, which went into effect in January 2016. The Canadian Institutes of Health Research [88] followed by requiring applicants to specifically integrate sex and gender into experimental design. While the European Commission has had a long-standing policy to question when sex and gender are relevant in the objectives and methodologies of a project, it hasn’t included a reporting requirement like the NIH or CIHR. More recently, the UK Research and Innovation Medical Research Council released their new guidance [89] that requires the specification of sex in the experimental design, effective September 2022. 

### 5.3. Accountability in Reporting Practices

There is a long-standing need to improve the reporting of experimental methods and materials [25]. To improve the quality and utility of animal-based study results, it has been previously recommended that journals and funding agencies mandate that reporting of animal studies include complete descriptions of all experimental details [90]. In 2010, a working group sponsored by the National Centre for the Replacement, Refinement and Reduction of Animals in Research (NC3Rs) published the ARRIVE Guidelines [83]. The purpose was to improve transparency in research reporting to help address the reproducibility crisis. Improved reporting of animal sex was included in these guidelines. While many journals have endorsed these guidelines, compliance and enforcement has been poor, and there continues to be incomplete reporting [84].

Building on this need for improved accountability in reporting practices, sex has been specifically identified as a variable to report. The SAGER (Sex and gender equity in research) guidelines were released in 2016 [91]. These guidelines provide a comprehensive approach to reporting sex and gender information in study design, data analysis, results, and interpretation of findings. While the SAGER guidelines are primarily designed for and by scientific writers, they are also useful to reviewers and editors to help ask questions such as whether sex is relevant to the research in question and/or have the authors adequately addressed sex-based effects or justified the absence of such analysis.

In 2020, after 10 years in practice, the ARRIVE guidelines were updated and reorganized to facilitate their use and renamed as ARRIVE 2.0 [84,92]. These guidelines specifically described sex as a property of the sample, and an independent variable that potentially affects the outcome measures. The authors acknowledged that sex effects can be accounted for in the randomization or blocking strategy and that including sex as a variable can increase power, thereby increasing the ability to detect a real effect with fewer animals. The ARRIVE 2.0 guidelines list animal details, including sex, as item 8 of the Essential 10 minimum reporting requirements [84]. 

While having guidelines and checklists that are endorsed by scientific journals is a good starting point, it requires effort from all members of the scientific community to create a culture of accountability in scientific reporting. Using guidelines such as SAGER and ARRIVE 2.0 makes it easy to report important experimental variables, but consistent use of the guidelines can be difficult without behavior modification by all parties. It isn’t sufficient for journals to simply endorse the guidelines; the guidelines must be fully adopted across all roles (editors, reviewers, and authors) and integrated into standard writing practices. Only together, through concerted efforts at the funding agency, institution, and publishing levels will the consideration of sex as a biological variable become standard practice in biomedical research [93]. 

### 5.4. Legislation

Using legislation to achieve sex balance of research animals is potentially an extreme approach to achieving the goal, but it is not unheard of. As mentioned previously, current German animal welfare legislation does not allow for killing of animals without a reasonable cause [94]. As a result, there have been criminal complaints filed against at least 15 biomedical research facilities for euthanasia of surplus animals [54]. While criminal charges have not been made to date, this highlights the importance of cooperative engagement and adoption of guidelines by the entire scientific community to voluntarily address sex as a biological variable to avoid such drastic measures.

### 5.5. Technology

Using a single sex is legitimately needed for some studies. In these cases, thoughtful uses of technology can be paramount to minimizing needless animal waste resulting from overproduction. Using sexed semen is commonplace in some fields of veterinary medicine [95,96,97,98]. Sexed semen has been used in the beef and dairy industries since 1989 to minimize production of select sexes with a high level of success [95], and this technology has been expanded to use with pigs, horses, and small ruminants [99,100,101]. A pilot study in Ireland demonstrated the welfare benefits of using sexed semen to reduce unwanted production of surplus male dairy calves [102]. Although not without some financial costs, use of this technology could occur for common laboratory species to minimize production of the particular unwanted sex and decrease surplus animal creation. Similarly, CRISPR-Cas 9 technology can be used to limit the in utero development of fetuses of a select sex [103]. Production of sex-specific offspring can become a heritable trait, making it easier to continue production of single sex litters in subsequent rounds of breeding. Using these technologies could significantly decrease the number of animals being euthanized due to overproduction, thus improving welfare for research animals. However, careful evaluation of single sex litters for unanticipated effects of genetic manipulation must be conducted and reported as intrauterine position and the sex of adjacent fetuses in utero have well-documented effects on a number of traits and behaviors later in life [104,105,106,107].

## 6. Conclusions

Sex bias in biomedical studies is bad for both scientific advancement and animal welfare. Not only has sex bias led to large financial losses for the pharmaceutical industry and harms for women as patients, but there are also significant welfare impacts for animals. There is currently a need to study both sexes in multiple research disciplines and to recognize when sex may or may not be an experimental variable. Ultimately, it is critically important that sex be considered at each stage of the scientific process. Additionally, using technology to minimize animal waste when a single sex is justified is important to ensure positive animal welfare. Collectively, these actions will improve both reproducibility and translatability to propel scientific discovery and therapeutic success while promoting animal welfare.

## Figures and Tables

**Figure 1 animals-13-02792-f001:**
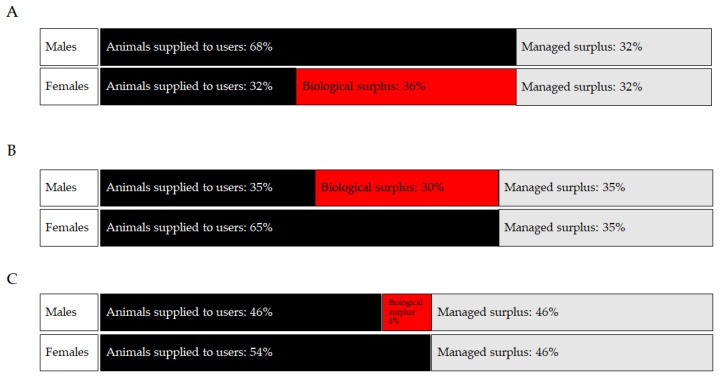
Use and surplus rats and mice by sex. In this example, the animals used for scientific procedures are represented by black bars, the excess animals created but not used (Managed surplus) are represented by grey bars, and the excess animals created but not used due to sex bias (Biological surplus) are represented by red bars. (**A**) Rats used and surplus animals. (**B**) Mice (inbred and outbred) used and surplus animals. (**C**) Mice (genetically modified) used and surplus animals. Adapted from [60].

**Table 1 animals-13-02792-t001:** Primary issues related to overproduction of research rodents (adapted from [60]).

Issue Resulting in Overproduction of Rodents	Definition of Issue
Sex preference	Use of one sex in preference to equal use of both sexes. Remainder of lesser used sex becomes surplus.
Body weight and age requirement	Requirement for a narrow age and/or body weight requirement, such that any animals outside of these requirements become surplus (any time up to study start).
Breeding pressures	Variable demand for animals of multiple strains, for example, changes in week-to-week orders, as well as short notice orders for animals. Creates pressures to have large numbers of animals ready at a given moment.
Timed mating	Inexact procedure such that more animals are mated than are needed. Surplus also created if study is cancelled after mating has occurred.
Poor health status	Animals with clinical disease may need to be euthanized, and more animals are bred to account for these losses. In addition, a large surplus may need to be produced if mouse strains need to be replaced or rederived because of unwanted colony infections (clinical or subclinical).
Part use of litter	Selective use of animals in a litter with remainder as surplus.
Study cancellation	Sudden cancellation of studies such that it may not be possible to reallocate animals to a different study within a given institution.
Genetics of breeding	Proportion of animals may not have the required genotype, and some animals may not be of appropriate quality, e.g., stunted or malformed.
Historical use	Use of specific strains that are no longer in common use because of historical database
Duplication of animal colonies	Duplication of in-house breeding colonies of multiple strains of mice and rats in large institutions
Ineffective management practices	Failure to manage breeding colonies efficiently, for example, by maintaining breeding at low levels when future use is uncertain.

**Table 2 animals-13-02792-t002:** A guide for incorporating sex as a biological variable in animal research studies (adapted from [85]).

Study Aspect	Actionable Items
Literature search	Perform literature search for sex difference in research area of interest Critically evaluate the literature for quality of reference to identify sex differences Understand the strengths and limitations of available literature
Study design	Use both sexes of animals unless scientifically justifiable Design studies with sufficient power to determine differences between experimental groups and sexes Best practice: factorial design with the sample size split between males and females
Data collection	Collect data from a similar number of animals of both sexes Collect data on variables that could influence sex differences (e.g., housing differences)
Data analysis	Separate male and female data sets and specifically test for sex differences
Reporting	Describe study design Report numbers, ages, and weights of males and females used Report sex differences tested for and identified (or not)

## Data Availability

No new data were created.

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
