# Peer review of "Unmasking the Adverse Impacts of Sex Bias on Science and Research Animal Welfare"

_animals, 2023, doi:10.3390/ani13172792_

Round 1

Reviewer 1 Report

The review article by Elizabeth A. Nunamaker and Patricia V. Turner accounts for a very common practice in biomedical research which is the sex bias in the use of one sex over the other in basic and preclinical studies. Although funding agencies from many countries are requiring both sexes to be studied in biomedical research, single sex studies are still common. This is a very important issue taking into account the vast and growing body of evidence about sex differences not only in physiology but also in cell processes and signaling pathways. Here, the authors review some commonly used justifications for not using both sexes and explore the implications for animal welfare. The quality of the English is high and the manuscript is overall well designed and written. I have some minor suggestions on possible typos (see below).

L151:”utsing” should be “using

L301 and 303: What authors mean by NSAID? Please clarify or revise text in case it is a typo

L314: mice are mentioned twice in this sentence.

L435: "form another institution" should be  "from another institution"

Author Response

Response to reviewers.

We thank the reviewers for their review and thoughtful comments that will contribute to the strength of this review. We have made the requested modifications as briefly described below and as highlighted in yellow in the manuscript.

Reviewer 1:

  • L151:”utsing” should be “using
    • Corrected
  • L301 and 303: What authors mean by NSAID? Please clarify or revise text in case it is a typo
    • We have spelled out NSAID – nonsteroidal anti-inflammatory drug in the text the first time this abbreviate is used on line 301
  • L314: mice are mentioned twice in this sentence.
    • Removed second reference to mice
  • L435: "formanother institution" should be  "from another institution"
    • Corrected

Reviewer 2 Report

The goals of this review article are to discuss how sex bias has developed over the past 100 years, the current status of sex bias, ongoing efforts to address sex bias, implications of these ongoing efforts, and suggest to additional methods to minimize sex bias in future biomedical studies. The manuscript is well-written, the topic and content are important, and will add to the limited research in this area. This should be of particular interest to the broad readership of the journal. Comments are mostly for clarity and quality improvement.

One of the most common misconceptions on reporting sex and gender differences is that negative results (findings of no sex differences) need not be reported. There is a need to educate here and explicitly state this should not discourage reporting. This is a major contributor to the wastage of animals as experiments need to be redone in both sexes. This point should be expanded on.

An important point that was not discussed in the article is the historical lack of reporting of the sex of animals and cell lines used. The authors note that sometimes sexes are unknown but historically animal or cell line sex might be known but still not reported perhaps due to the notion that it was not important to report. This makes it challenging to identify the generalizability of the study and creates a need to redo studies.

It is important to draw attention to the SAGER guidelines as an established systematic approach to reporting sex and gender in research across disciplines.

Heidari, S., Babor, T.F., De Castro, P. et al. Sex and Gender Equity in Research: rationale for the SAGER guidelines and recommended use. Res Integr Peer Rev 1, 2 (2016). https://doi.org/10.1186/s41073-016-0007-6

“Successful incorporation of hormone variation in study design…” is an important topic and needs to be expanded on. While the authors cite important literature for reference, this is a need to educate here and provide more details so that is seems attainable to the readers and to encourage prospective as opposed to retrospective incorporation of sex and gender into study designs and data analyses.

Authors should expand on the point regarding justification for including one sex “when the reasoning is scientifically sound” with specific examples of when it is and isn’t justifiable.

The following sentence needs citations. “…there is evidence to suggest that sex bias results in an overproduction of research animals, inadequate pain management, and results in significant animal wastage”

Include the year/first grant cycle in which the NIH implemented its policy requiring the inclusion of both sexes or strong justification otherwise.

The idea that sex bias has contributed to challenges in identifying pain and pain management is alluded to several times throughout the paper without explanation and citation until page 7. Please direct the reader to this section and/or expand on this briefly sooner.

Typo: “utsing”

Figure 1: Label A, B, and C with rats, mice and genetically modified mice, respectively.

More context is needed for CYP2D6 and Cyp3A4 in the role of pain.

NC3Rs and ARRIVE are discussed before they are defined and explained in the text.

Author Response

Response to reviewers.

We thank the reviewers for their review and thoughtful comments that will contribute to the strength of this review. We have made the requested modifications as briefly described below and as highlighted in yellow in the manuscript.

Reviewer 2:

  • One of the most common misconceptions on reporting sex and gender differences is that negative results (findings of no sex differences) need not be reported. There is a need to educate here and explicitly state this should not discourage reporting. This is a major contributor to the wastage of animals as experiments need to be redone in both sexes. This point should be expanded on.
    • A paragraph has been added to describe the detrimental effects of this practice and its contributions to animal wastage in section 4.4.
  • An important point that was not discussed in the article is the historical lack of reporting of the sex of animals and cell lines used. The authors note that sometimes sexes are unknown but historically animal or cell line sex might be known but still not reported perhaps due to the notion that it was not important to report. This makes it challenging to identify the generalizability of the study and creates a need to redo studies.
    • A sentence about this practice has been added on Lines 98-100.
  • It is important to draw attention to the SAGER guidelines as an established systematic approach to reporting sex and gender in research across disciplines.
  • Heidari, S., Babor, T.F., De Castro, P. et al.Sex and Gender Equity in Research: rationale for the SAGER guidelines and recommended use. Res Integr Peer Rev 1, 2 (2016). https://doi.org/10.1186/s41073-016-0007-6
    • We have added a paragraph about the SAGER guidelines and have added the recommended citation to the manuscript.
  • “Successful incorporation of hormone variation in study design…” is an important topic and needs to be expanded on. While the authors cite important literature for reference, this is a need to educate here and provide more details so that is seems attainable to the readers and to encourage prospective as opposed to retrospective incorporation of sex and gender into study designs and data analyses.
    • A paragraph describing a practical example has been added to Section 3.2 to help illustrate how hormone variation can be incorporated into study design.
  • Authors should expand on the point regarding justification for including one sex “when the reasoning is scientifically sound” with specific examples of when it is and isn’t justifiable.
    • Examples of reasonable and questionable justifications have been added to section 3.3.
  • The following sentence needs citations. “…there is evidence to suggest that sex bias results in an overproduction of research animals, inadequate pain management, and results in significant animal wastage”
    • Line 57 – rather than add all of the citations, we have directed the reader to the respective sections of the manuscript for the supporting information.
  • Include the year/first grant cycle in which the NIH implemented its policy requiring the inclusion of both sexes or strong justification otherwise.
    • Line 446 - January 2016 was added
  • The idea that sex bias has contributed to challenges in identifying pain and pain management is alluded to several times throughout the paper without explanation and citation until page 7. Please direct the reader to this section and/or expand on this briefly sooner.
    • Line 57 – we have directed the reader to section 4.2 for the supporting information.
  • Typo: “utsing”
    • Corrected
  • Figure 1: Label A, B, and C with rats, mice and genetically modified mice, respectively.
    • Thank-you - these are identified in the text of the figure header
  • More context is needed for CYP2D6 and Cyp3A4 in the role of pain.
    • An additional sentence was added to provide context
  • NC3Rs and ARRIVE are discussed before they are defined and explained in the text.
    • ARRIVE is now defined on line 380-381, the first time it appears.